# Cryo-EM structure of the adenosine A$_{2A}$ receptor coupled to an engineered heterotrimeric G protein

Javier García-Nafría[†], Yang Lee[†], Xiaochen Bai[‡], Byron Carpenter[§], Christopher G Tate*

MRC Laboratory of Molecular Biology, Cambridge, United Kingdom

**Abstract** The adenosine A$_{2A}$ receptor (A$_{2A}$R) is a prototypical G protein-coupled receptor (GPCR) that couples to the heterotrimeric G protein G$_S$. Here, we determine the structure by electron cryo-microscopy (cryo-EM) of A$_{2A}$R at pH 7.5 bound to the small molecule agonist NECA and coupled to an engineered heterotrimeric G protein, which contains mini-G$_S$, the βγ subunits and nanobody Nb35. Most regions of the complex have a resolution of ~3.8 Å or better. Comparison with the 3.4 Å resolution crystal structure shows that the receptor and mini-G$_S$ are virtually identical and that the density of the side chains and ligand are of comparable quality. However, the cryo-EM density map also indicates regions that are flexible in comparison to the crystal structures, which unexpectedly includes regions in the ligand binding pocket. In addition, an interaction between intracellular loop 1 of the receptor and the β subunit of the G protein was observed.

DOI: https://doi.org/10.7554/eLife.35946.001

*For correspondence:
cgt@mrc-lmb.cam.ac.uk

[†]These authors contributed equally to this work

Present address: [‡]Department of Biophysics, University of Texas Southwestern Medical Center Dallas, Dallas, United States; [§]Warwick Integrative Synthetic Biology Centre, University of Warwick, Warwick, United Kingdom

## Introduction

The adenosine A$_{2A}$ receptor (A$_{2A}$R) is an archetypical Class A G-protein-coupled receptor (GPCR) (*Venkatakrishnan et al., 2013*). A$_{2A}$R is activated by the endogenous agonist adenosine and plays a prominent role in cardiac function, the immune system and central nervous system, including the release of the major excitatory neurotransmitter glutamate (*Fredholm et al., 2001*; *Fredholm et al., 2011*). Given the widespread tissue distribution and physiological relevance of A$_{2A}$R, it is a validated drug target for many disorders (*de Lera Ruiz et al., 2014*), including Parkinson's disease (*Hickey and Stacy, 2012*) and cancer (*Leone et al., 2015*). A$_{2A}$R is one of the most stable GPCRs and structures have been determined of A$_{2A}$R in an inactive state bound to inverse agonists (*Doré et al., 2011*; *Jaakola et al., 2008*; *Congreve et al., 2012*; *Hino et al., 2012*; *Liu et al., 2012*; *Segala et al., 2016*; *Sun et al., 2017*), an active intermediate state bound to agonists (*Lebon et al., 2015*; *Lebon et al., 2011*; *Xu et al., 2011*) and the fully active state bound to an agonist and coupled to an engineered G protein, mini-G$_S$ (*Carpenter et al., 2016*). In addition, structure-based drug design has been applied to inactive state structures of A$_{2A}$R to develop potent and subtype specific inverse agonists with novel scaffolds (*Congreve et al., 2012*) and these are currently in clinical trials. Comparison of the structures has led to an understanding of the molecular determinants for an inverse agonist compared to an agonist (*Lebon et al., 2011*), the conformational changes induced by agonist binding to convert the inactive state to the active intermediate state (*Lebon et al., 2012*), and the role of the G protein in stabilising the fully active state (*Carpenter et al., 2016*). The active state was determined by crystallizing the receptor coupled solely to mini-G$_S$, an engineered G protein with eight point mutations and three deletions, including the whole of the α-helical domain (*Carpenter and Tate, 2016*). Although pharmacologically mini-G$_S$ recapitulates the ability of a heterotrimeric G protein to increase the affinity of agonist binding to

the receptor (*Carpenter et al., 2016*), the roles for the βγ subunits could not be described. In terms of the interactions between a heterotrimeric G protein and a Class A GPCR, the vast majority of interactions are made by the α subunit, in particular the C-terminal α5 helix (*Rasmussen et al., 2011*). However, there was an interaction between the β subunit and the β2-adrenoceptor (*Rasmussen et al., 2011*) and also between the β subunit and the class B receptors for calcitonin (*Liang et al., 2017*) and glucagon-like peptide (*Zhang et al., 2017*; *Liang et al., 2018*). In addition, there is mutagenesis data suggesting that the α2-adrenergic receptor directly interacts with the β subunit (*Taylor et al., 1994*; *Taylor et al., 1996*). We therefore determined the structure of A2AR coupled to an engineered heterotrimeric G protein.

There are now two choices in how to determine the structure of a GPCR coupled to a heterotrimeric G protein, which are X-ray crystallography and electron cryo-microscopy (cryo-EM). The disadvantage of X-ray crystallography lies in the difficulty of producing good quality crystals of a GPCR coupled to a heterotrimeric G protein. The only successful strategy so far has been to use lipidic cubic phase composed of the lipid MAG7:7 and to crystallise a GPCR fusion protein with T4 lysozyme at the N-terminus, but there is only a single structure published to date (*Rasmussen et al., 2011*). The other option is to use cryo-EM and single particle reconstruction techniques. This is now possible given the recent developments in the field over the last few years (*Fernandez-Leiro and Scheres, 2016*) together with the improved contrast provided by the recently developed Volta phase plate (VPP) (*Danev et al., 2014*), which enhances the probability of getting structural data of small proteins (*Khoshouei et al., 2017*). The recent structure determination of two Class B receptors coupled to G$_S$ also shows the potential of this methodology (*Liang et al., 2017*; *Zhang et al., 2017*; *Liang et al., 2018*). We thus decided to use cryo-EM to determine the structure of A2AR coupled to an engineered heterotrimeric G protein. This would provide insights about the role of the β subunit in coupling to A2AR, but would also provide an opportunity to directly compare the structure of the receptor determined in the active state by X-ray crystallography and cryo-EM.

## Results

### Preparation of an A$_{2A}$R-G$_S$ complex

In this work, we used a construct of A2AR that contained thioredoxin at the N-terminus of the receptor (*Nehmé et al., 2017*). This was originally designed with a rigid linker between the thioredoxin and the receptor to generate a large hydrophilic surface to A2AR to improve crystallisation, although this proved unsuccessful. The presence of thioredoxin did not significantly affect the pharmacology of A2AR, as assessed by determination of its apparent K$_D$ for the inverse agonist ZM241385 or in agonist shift assays (*Figure 1*). It could also be purified to homogeneity and coupled effectively to both mini-G$_S$ (*Nehmé et al., 2017*) and to the heterotrimer containing mini-G$_S$, β$_1$, γ$_2$ and Nb35 (*Figure 1*). Detergent-solubilised A2AR coupled to the heterotrimer had a molecular weight (excluding the detergent micelle of LMNG) of approximately 135 kDa (*Nehmé et al., 2017*).

### The impact of the Volta Phase Plate on the cryo-EM A$_{2A}$R-G-protein complex map

Initial micrographs for the A2AR complex were collected on a FEI Titan Krios microscope using a K2 Summit detector in the absence of a Volta-potential phase plate (VPP) (*Figure 2a*). Data processing showed the characteristic 2D class averages of a GPCR coupled to a heterotrimeric G protein (*Figure 2a*). After 3D classification and refinement, the best model (containing 72,486 particles) reached 6.7 Å resolution and showed clearly defined α-helices in both the receptor and G protein (*Figure 2a*). We then collected data using the VPP on a FEI Titan Krios microscope using either a K2 Summit detector or a Falcon III detector in electron counting mode (*Figure 2b and c*). The K2 dataset consisted of micrographs pooled from different days and collected with slight variations regarding total dose and doses rates (see Materials and methods for details), while the Falcon III dataset was collected in a single session over 48 hr. Both datasets were processed in an equivalent manner to the non-VPP data, with only few minor exceptions (see Materials and methods). Since images collected with a VPP possess higher contrast (*Figure 2b and c*), the auto-picking feature in RELION that uses a Gaussian blob as a reference resulted in optimal particle picking without the need for specific 'auto-picking' references (*Fernandez-Leiro and Scheres, 2017*). After 2D and 3D

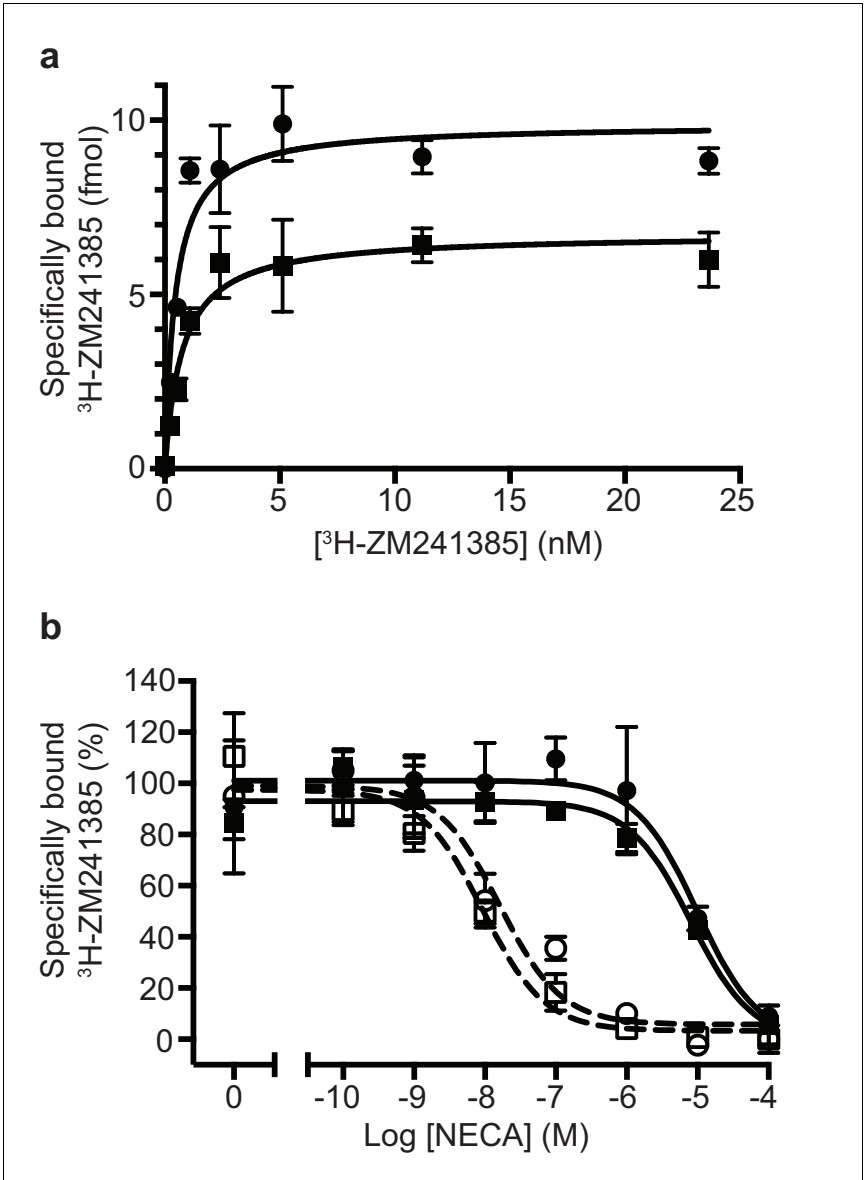

**Figure 1.** Pharmacological analyses of $A_{2A}R$. (a) Saturation binding of the inverse agonist $^3$H-ZM241385 to $A_{2A}R$ constructs gave the following apparent $K_D$s: $A_{2A}R$ (circles), 0.5 ± 0.1 nM; TrxA-$A_{2A}R$ (squares), 0.8 ± 0.2 nM. (b) Competition binding curves measuring the displacement of $^3$H-ZM241385 with increasing concentrations of NECA gave the following $K_i$s for NECA; $A_{2A}R$ (filled circles), 1.0 ± 0.5 μM; $A_{2A}R$ + mini-$G_S$ (open circles, dashed line), 2.6 ± 1.8 nM; TrxA-$A_{2A}R$ (filled squares), 1.1 ± 0.4 μM; TrxA-$A_{2A}R$ + mini-$G_S$ (open squares, dashed line), 1.8 ± 1.2 nM. Data plotted are the average from two independent experiments performed in duplicate with error bars shown as the SD.

DOI: https://doi.org/10.7554/eLife.35946.002

The following source data is available for figure 1:

**Source data 1.** Raw data for A2aR competition binding curves.

DOI: https://doi.org/10.7554/eLife.35946.003

classification (see Materials and methods for details), refinement yielded models with overall resolution of 4.88 Å and 4.45 Å for the K2 Summit and Falcon III detector, respectively (**Figure 2b and c**). The Falcon III model was later improved to 4.11 Å with further processing (see below) showing details for most amino acid side chains after B factor sharpening. The effect of the VPP for this particular dataset was therefore essential to make 'side-chain' resolution accessible. A B-factor plot

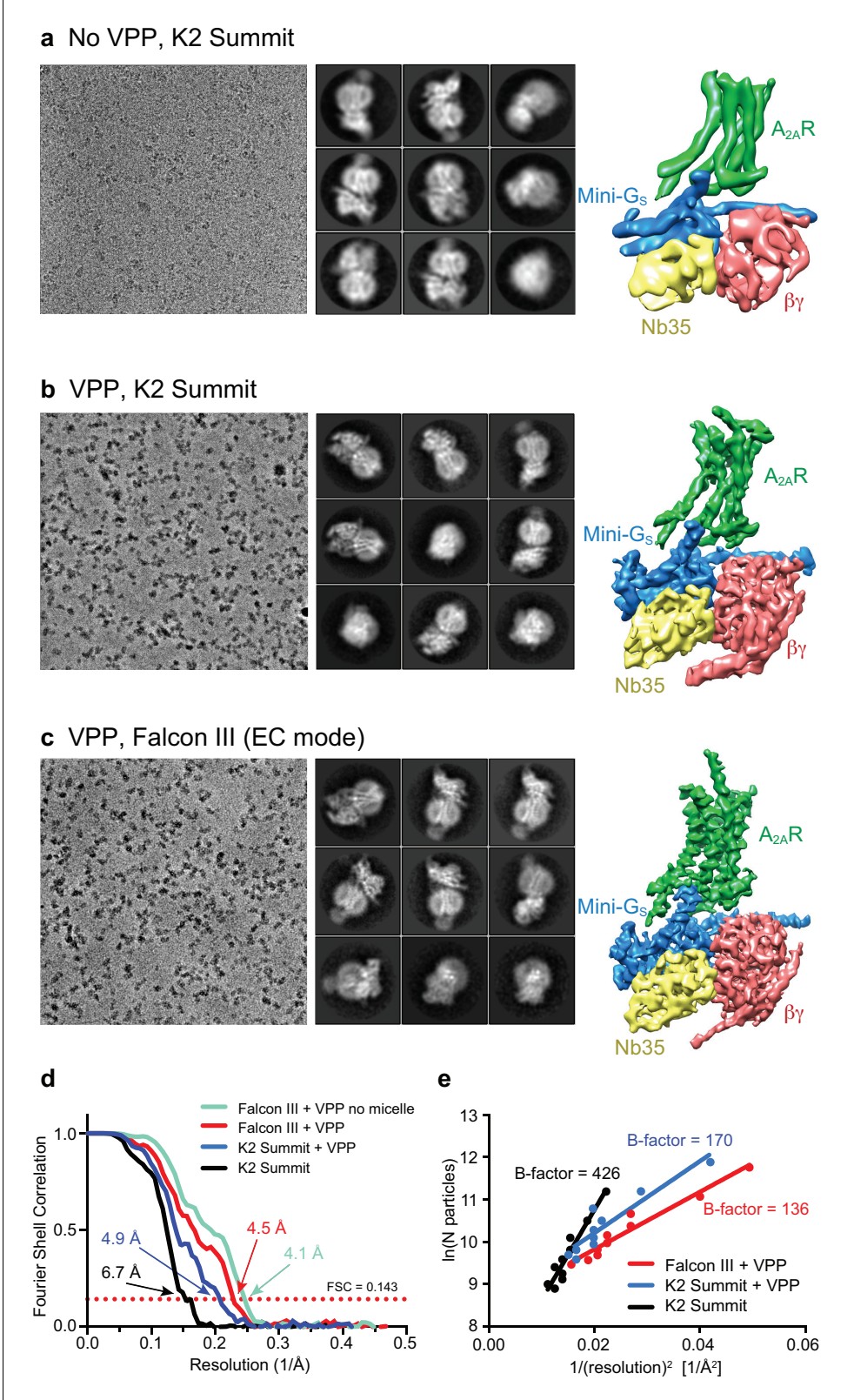

**Figure 2.** Cryo-EM of the $A_{2A}R$ complex in the presence and absence of a VPP. (**a-c**) Each panel contains three sections, with the left-hand section showing a representative micrograph obtained on a Titan Krios, the central section depicting 2D class averages and the right-hand section the refined 3D reconstruction obtained from the data collected. (**a**) Data collected without using a VPP on a K2 Summit detector. (**b**) Data collected using a VPP on

*Figure 2 continued on next page*

*Figure 2 continued*

a K2 Summit detector. (**c**) Data collected using a VPP on a Falcon III detector in electron counting (EC) mode. (**d**) Gold-standard FSC curves for the three 3D reconstructions with resolutions estimated at 0.143. (**e**) Difference in B-factors between the three datasets.

DOI: https://doi.org/10.7554/eLife.35946.004

(assessing the number of particles vs resolution) was used to assess the impact of the VPP (*Figure 2e*). It is observed that, in the presence of the VPP, the $A_{2A}R$ map not only has a better resolution for the same number of particles, but the B-factor improves significantly from 426 to 170 (when comparing K2 Summit with and without VPP). This becomes essential when trying to reach high-resolution information in a reasonable time scale (especially important for high-throughput structure determination in drug discovery). As an example, to obtain the same resolution of 4.88 Å using the K2 Summit detector without the VPP, one would have needed about 5 million particles, that would require ~65 days of data collection at a Titan Krios electron microscope (in comparison to 145,169 particles collected in 48 hr with a VPP).

All cryo-EM grids were plunge-frozen from a single batch of $A_{2A}R$–G protein complex and most of the duplicate grids were made in a single freezing session. Data collection was performed at higher magnification for the non-VPP data (magnification 200,000x and 0.66 Å/pixel) than for the VPP dataset (1.14 Å/pixel and 1.07 Å/pixel for the K2 Summit and Falcon III detectors, respectively), positioning the high-resolution information of the non-VPP data at a better location in the detector DQE range (Nyquist being 1.32 Å vs 2.14 Å/2.28 Å for the non-VPP vs the VPP K2/Falcon III, respectively). The VPP resolution enhancement therefore could potentially be higher if equivalent magnifications were used. Data processing was carried out as equivalent as possible for all datasets in order to make them comparable. We therefore believe that the comparison between the VPP and non-VPP datasets is as fair as possible, although if anything we are favouring the non-VPP data.

Comparisons of data with and without VPP had only been previously been published for samples that readily reached high resolution without VPP. Although in our experience, the improvement is sample dependent, these data show the potential to which the VPP can be useful in certain cases and more comparisons will be needed in order to understand the variability in enhancement between samples. Although we see a significant difference between the K2 and Falcon III performance, data for the K2 with VPP was a result of merging data with different dose rates and total doses. We therefore do not have an absolutely identical comparison of the two detectors.

## Structure determination of the $A_{2A}R$–$G_S$ complex

The highest resolution data set corresponded to micrographs collected on a Falcon III detector in electron counting mode using a VPP, therefore this map was used for further processing, model building and subsequent analysis. Data collection parameters and processing are described in the Materials and methods section. In summary, 837 movies were collected and corrected for stage drift, beam induced movement and dose weighting with MotionCor2 (*Zheng et al., 2017*). CTF fitting, defocus and phase estimation were performed with Gctf-v0.1.06 (*Zhang, 2016*). Particle picking was performed using a Gaussian blob, as implemented by RELION (*Scheres, 2012*). 3D classification was performed with an *ab initio* model and refinement of the best classes with clear GPCR-like features (128,002 particles) attained an overall resolution of 4.45 Å (using gold standard FSC of 0.143) (*Rosenthal and Henderson, 2003*). Attempts to improve the model included further 3D classification, which revealed that around 50% of the particles contained a heterogeneous γ subunit. However, the resolution and quality of the overall model suffered when removing these particles, so we therefore compromised on having poor quality density for the γ subunit, but having higher resolution for the rest of the complex.

In further attempts to improve the model, during refinement, the low-pass filter effect of the Wiener filter in the regularised likelihood optimisation algorithm was relaxed through the use of a regularisation parameter (T = 5). This allowed the refinement algorithm to consider higher spatial frequencies in the alignment of the individual particles yielding a map of higher quality. Nevertheless, both half-reconstructions were kept completely separately, and the final resolution estimate (at the post-processing stage in RELION) was based on the standard FSC between the two unfiltered

half-reconstructions. Although resolution did not improve, the quality of the map improved noticeably.

Calculation of the local resolution in RELION showed that although the overall resolution was estimated to be 4.5 Å the core of the complex was ~3.8 Å with most of the map at 4.0 Å resolution or higher, with clearly visible density for the majority of amino acid side chains. As shown in *Figure 3*, the regions that showed poorer resolution were the thioredoxin and the detergent micelle (a significant fraction of the small complex), which hinders a realistic overall resolution estimation.

In order to accurately estimate the resolution of the $A_{2A}R$ complex map and to eliminate noise from refinement, the detergent micelle and thioredoxin moiety needed to be excluded. Excluding the micelle by simply tightening the mask did not yield optimal results with artefacts produced at the interface between the model and the mask. Such a strong signal might be specific to LMNG since, in our experience, the signal from other detergents can be masked out in this manner. We

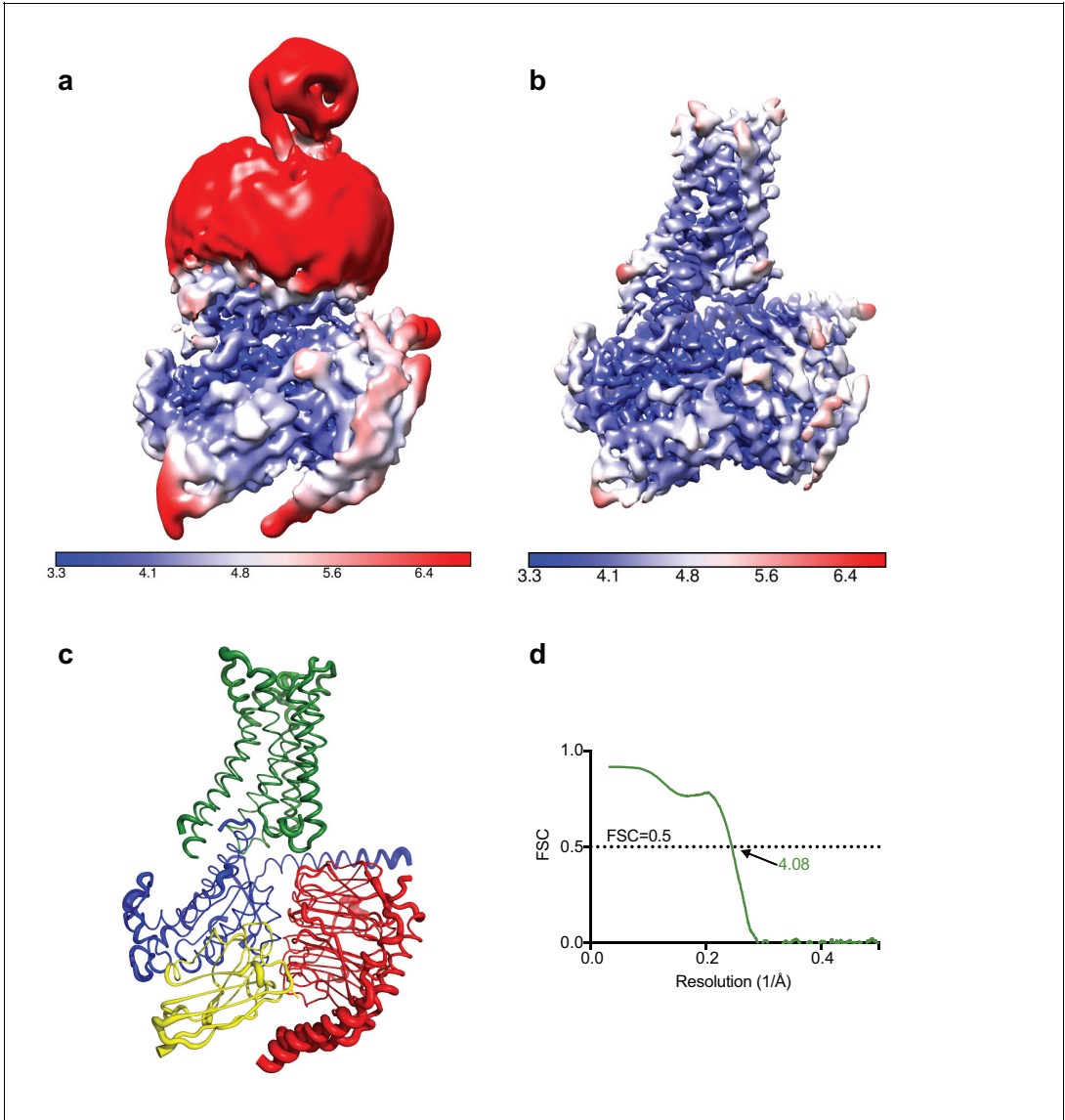

**Figure 3.** Local resolution cryo-EM map. (a) Local resolution map of the Falcon III + VPP model prior to refinement with signal subtracted particles as calculated with RELION. (b) Local resolution of the same model after refinement of signal subtracted particles (also calculated with RELION) (c) $A_{2A}R$ complex displayed as putty cartoons, where B-factor of the coordinates relates to the thickness of the tube. (d) Fourier shell correlation of the refined model versus the map.

DOI: https://doi.org/10.7554/eLife.35946.005

then decided to perform a double signal subtraction protocol where initial coordinates were used to create a tight mask around the protein component excluding thioredoxin ($A_{2A}R$, mini-$G_S$, β, γ, Nb35), which was then subtracted from the original particles. The resulting particles were used to produce an accurate map of the micelle and thioredoxin, which was then used to perform signal subtraction of the original particles, leaving them devoid of micelle or thioredoxin. Refinement of these particles yielded an improved map at 4.11 Å resolution. However, it appeared that the refinement process focused primarily on the intracellular G protein heterotrimer complex leaving a lower quality map at the receptor region. In order to circumvent this problem, we performed refinement with the original particles and then exchanging them for their signal subtracted equivalent (without micelle and thioredoxin) only in the last iteration of refinement. This resulted in the best overall map at 4.11 Å resolution with quality density throughout (*Table 1*).

Attempts to remove particles with low phase shift and poor contrast (~22,000 particles with <0.25π) decreased resolution and map quality. We therefore kept low phase shift data in the final model.

## Overall structure of the NECA-bound $A_{2A}R$ mini heterotrimeric G protein complex

The $A_{2A}R$ cryo-EM complex structure provides insights into its structure in solution, in the absence of crystal contacts and at more physiological conditions (pH 7.5) than the X-ray structure (pH 5.7 or below for inactive structures). The density map of the $A_{2A}R$–G protein heterotrimer displayed a local resolution varying from 3.3 Å to 6.4 Å (*Figure 3*). Side chain densities were observed for most amino acid residues (*Figure 4*), which were of similar quality to those in the X-ray crystallographic map of the $A_{2A}R$–mini-$G_S$ structure (*Figure 5*). The lowest resolution was found at the C-terminus of the β subunit and most of the γ subunit, which had very poor density. Signal subtraction and 3D classification protocols have been used to isolate different protein conformations of small regions (*Bai et al., 2015*). Upon implementation of these strategies, we did not find any other discrete conformations of the heterotrimeric G protein, suggesting that the C-terminus of the β subunit and most of the γ subunit region are flexible. Within the cryo-EM structure of $A_{2A}R$, there are two regions that lack density and are therefore also probably disordered and flexible, namely the N-terminal section of ECL2 (G147 to Q163) and the whole of ICL3 (E212 to S223). These regions are ordered in some crystal structures, but this usually correlates with these regions forming lattice contacts. Sections of the cryo-EM density map for which there is poor quality density and high B-factors of the refined coordinates include TM1, helix 8, the second section of ECL2 that contributes relevant residues for ligand binding (see below), ECL1 and ECL3 (*Figures 3* and *4*).

The overall architecture of the $A_{2A}R$–heterotrimeric G protein complex is similar to the heterotrimeric $G_S$-coupled complexes for the $β_2$-adrenergic receptor (*Rasmussen et al., 2011*), GLP1 (*Zhang et al., 2017*, *Liang et al., 2018*) and the calcitonin receptor (*Liang et al., 2017*). The receptor and mini-Gs portions of the $A_{2A}R$–G protein complex are very similar to the crystal structure of the $A_{2A}R$–mini-$G_S$ complex, with the RMSD of $C_α$ atoms for the receptor and mini-$G_S$ components being 0.5 Å and 0.6 Å, respectively. The largest differences are found at the interface between mini-$G_S$ and the β subunit, which have a different conformation when βγ is bound. This may contribute to a minor difference in curvature of the α5 helix in mini-$G_S$ when it is in the heterotrimer complex compared to when it is bound to the receptor alone (*Figure 5*). However, this does not have any major impact on the interface between the receptor and mini-$G_S$, thus further validating the use of mini G proteins as a surrogate for G protein heterotrimers (*Carpenter et al., 2016*; *Carpenter and Tate, 2016*; *Nehmé et al., 2017*).

## Cryo-EM map at the ligand binding pocket

The $A_{2A}R$ orthosteric binding pocket is described by two crystal structures of $A_{2A}R$ bound to NECA, with one structure of $A_{2A}R$ in an active intermediate conformation (PDB code 2ydv) (*Lebon et al., 2011*) and the other structure in the active state coupled to mini-$G_S$ (PDB code 5g53) (*Carpenter et al., 2016*). The extracellular half of $A_{2A}R$ does not undergo any major structural changes in the transition from the active intermediate to the mini-$G_S$ coupled active state, with the volume of the binding pocket remaining constant and the interactions to NECA being identical (*Carpenter et al., 2016*). The orthosteric binding site in the cryo-EM map has well-defined density,

**Table 1.** Data collection and refinement statistics

**Data collection**

| Microscope | FEI titan krios | FEI titan krios | FEI titan krios |
|---|---|---|---|
| Detector | Falcon III + VPP | K2 Summit + VPP | K2 Summit |
| Pixel size (Å) | 1.07 | 1.14 | 0.66 |
| Voltage (kV) | 300 | 300 | 300 |
| Total electron dose (e⁻/Å²) | 30 | 40/40/30 | 50 |
| Micrographs collected | 827 | 906 | 2800 |
| Number of frames | 75 | 40/23/30 | 40 |
| Exposure time (s) | 60 | 10/4.6/6.5 | 10 |
| Electron dose per frame (e⁻/Å²) | 0.4 | 1/1.7/1 | 1.25 |
| Dose rate (e⁻/pixel/s) | 0.5 | 5.2/9/6 | 2 |
| Frame exposure (s) | 0.8 | 0.25/0.115/0.216 | 0.25 |
| Total number of particles (after 2D classification) | 232,739 | 313,879 | 166,313 |
| cryo-EM 3D Refinement | | | |
| Resolution (Å) | 4.11 | 4.88 | 6.71 |
| Map sharpening B-factor (Å²) | −130 | −150 | −529 |
| Fourier shell correlation criterion | 0.143 | 0.143 | 0.143 |
| Particles used in final 3D refinement | 128,002 | 145,169 | 72,487 |
| Defocus (μm) | −0.2 to −1 | −0.3 to −1.2 | −1.2 to −3.5 |
| Coordinate Refinement and Validation | | | |
| R.m.s. deviations | | | |
| Bonds (Å) | 0.07 | | |
| Angles (°) | 0.984 | | |
| Ramachandran Favoured (%) | 94.6 | | |
| Ramachandran Allowed (%) | 4.89 | | |
| Ramachandran Outliers (%) | 0.51 | | |
| Molprobity score | 1.36 | | |
| Clashcore, all atoms | 0.79 | | |
| Favoured rotamers | 91.12 | | |
| EMRinger score | 1.93 | | |
| FSC (model vs map - 0.5 cut-off) (Å) | 4.08 | | |
| PDB and map deposition | | | |
| PDB ID | 6GDG | | |
| EMDB ID | 4390 | | |

DOI: https://doi.org/10.7554/eLife.35946.006

although the map has lower resolution towards the extracellular surface. The density for NECA is of sufficient quality to allow an unambiguous orientation of NECA and the same interactions to the receptor are observed as present in the crystal structures (*Figures 5* and *6*).

Despite the similarities between the orthosteric binding site observed in the cryo-EM and X-ray structures, small differences were found in ECL2 that forms part of the binding pocket. The C-terminal half of ECL2 in the X-ray structures forms a helical turn that caps the pocket and contributes side chains that interact with NECA (Phe168 and Glu169). In the cryo-EM structure this region is more disordered. As a consequence, there is no significant side chain density for Phe168 and Glu169 in the cryo-EM map. The fact that there is clear density for NECA and His264 excludes the possibility that the whole of this region has poor resolution that is the extracellular portion of the receptor is not moving as a rigid body. This is consistent with ECL2 being dynamic.

**a**

```
7    AVYITVELAIAVLAILGNVLVCWAVWLNSNLQNVTNYFVVSLAAADIAVGVLAIPFAITI    66

67   STGFCAACHGCLFIACFVLVLTQSSIFSLLAIAIDRYIAIRIPLRYNGLVTGTRAKGIIA    126

127  ICWVLSFAIGLTPMLGWNNCGQPKEGKAHSQGCGEGQVACLFEDVVPMNYMVYFNFFACV    186

187  LVPLLLMLGVYLRIFLAARRQLKQMESQPLPGERARSTLQKEVHAAKSLAIIVGLFALCW    246

247  LPLHIINCFTFFCPDCSHAPLWLMYLAIVLSHTNSVVNPFIYAYRIREFRQTFRKIIRSH    306

307  VLRQQEPFKA    316
```

**b**

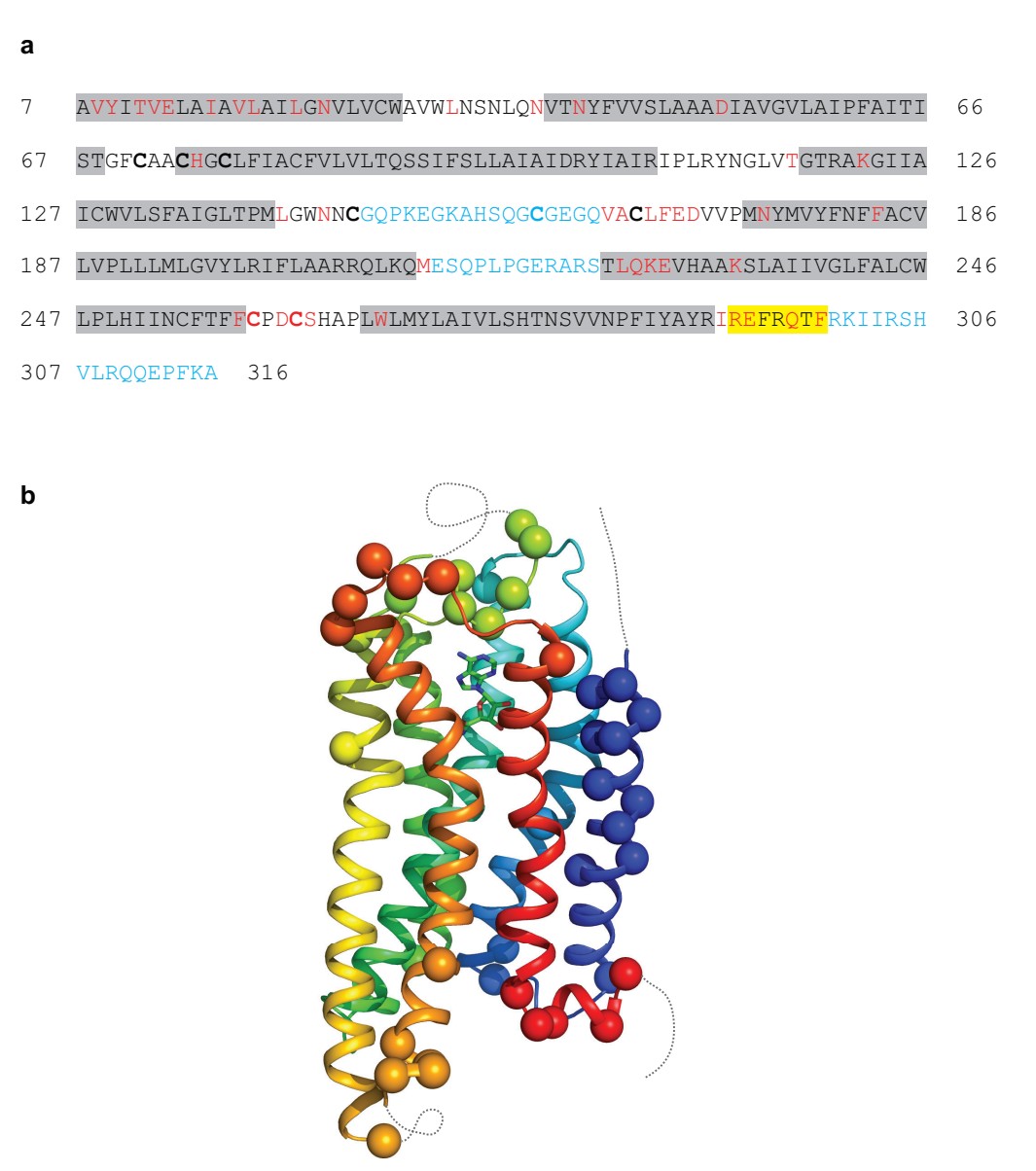

**Figure 4.** Modelling quality of the A$_{2A}$R structure. (a) Amino acid sequence of A$_{2A}$R used in the cryo-EM structure determination. Residues are coloured according to how they have been modelled: black, good density allows the side chain to be modelled; red, limited density for the side chain present and therefore the side chain has been truncated to Cβ; blue, no density observed and therefore the residue was not modelled. Regions highlighted in grey represent the transmembrane α-helices and amphipathic helix eight is highlighted in yellow. Cys residues involved in the formation of disulphide bonds are in bold. In the cryo-EM structure densities for the disulphide bonds Cys74-Cys146 and Cys77-Cys166 are observed. Densities corresponding to the disulphide bonds Cys71-Cys159 and Cys259-Cys262 are not observed in the cryo-EM data. The sequence of A$_{2A}$R is from residue 8–316, with the initial Ala residue at position seven being part of the linker between the N-terminal thioredoxin fusion and A$_{2A}$R. (b) Model of A$_{2A}$R showing the Cα positions of amino acid residues with poor density (spheres) and regions unmodelled (dotted lines).
DOI: https://doi.org/10.7554/eLife.35946.007

A second difference between the NECA-bound X-ray structures and the cryo-EM structure is a likely absence of an interaction between Glu169 and His264. This ionic bridge affects small molecule binding kinetics (*Segala et al., 2016*) and in most of the crystal structures caps the binding pocket. Although the cryo-EM map in the region is poorer than in the rest of the molecule, it suggests a rotamer for the imidazole group of His264 that points away from the orthosteric binding pocket (*Figure 6*). This might be a consequence of the pH in which the respective structures were determined.

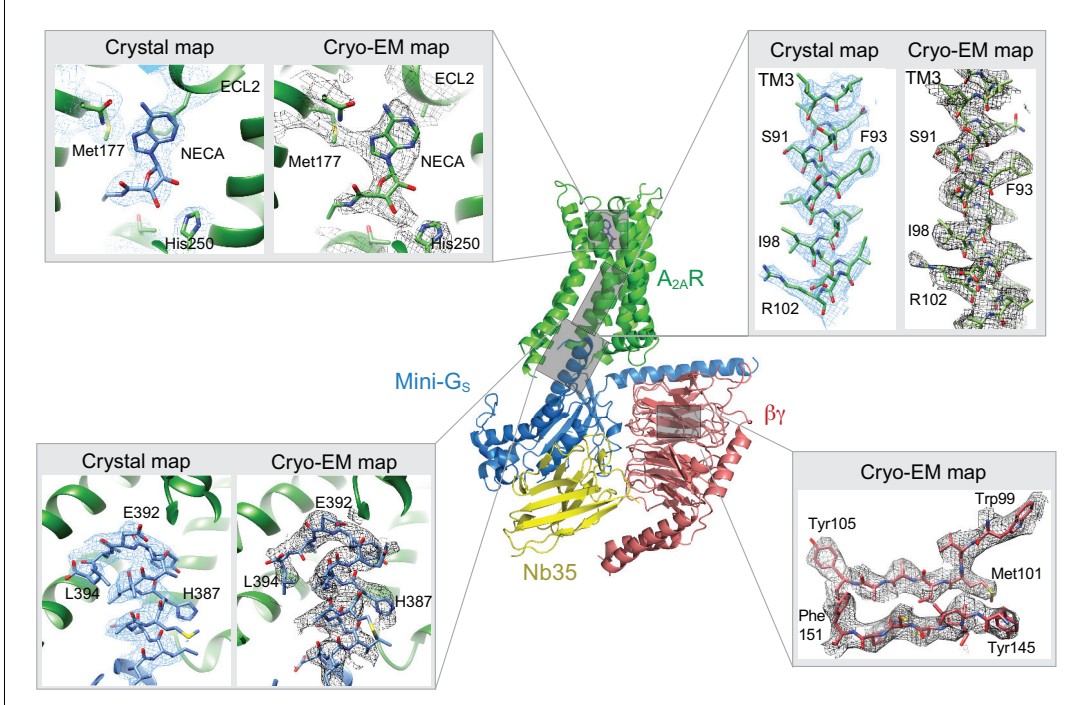

**Figure 5.** Comparison of map densities from the cryo-EM data and X-ray diffraction data. The structure of the A$_{2A}$R–heterotrimeric G protein complex determined by cryo-EM is depicted as a cartoon. The four panels show regions of the structure and the associated density maps from the cryo-EM data and, where present, electron density (2Fo-Fc) from the X-ray structure of the A$_{2A}$R–mini-G$_S$ (PDB code 5g53). Densities for the maps shown in the panels were sharpened using the following B factors (resolution of filtering in parentheses): β subunit and A$_{2A}$R, −170 Å$^2$ (3.7 Å); mini-G$_S$–A$_{2A}$R interface, −130 Å$^2$ (3.7 Å); NECA, −130 Å$^2$ (4.1 Å).

DOI: https://doi.org/10.7554/eLife.35946.008

The pK$_a$ of the histidine side chain is ~6 and most crystal structures have been obtained at lower pH (~pH 5), favouring protonation of His264 and the formation of the ionic bridge. At a more physiological pH of 7.5 that was used for the cryo-EM structure, His264 would be predominantly deprotonated and unable to form the ionic bridge. Although the rotamer for Glu169 cannot be assigned in the cryo-EM map, His264 adopts a similar rotamer as seen in crystal structures obtained at higher pH, such as the complexes with caffeine, XAC and ZM241385 (~pH 8), all showing a broken ionic bridge. Therefore, it is likely that in the physiological state (represented by the cryo-EM map) this ionic bridge is also absent, unless the surrounding pH is momentarily lowered for specific functions (e.g. the release of high concentrations of glutamate in glutamatergic synapses).

## Comparison of the G-protein–receptor interface between the crystal and cryo-EM structures

The interface between mini-G$_S$ in the heterotrimeric G protein and A$_{2A}$R in the cryo-EM structure is very similar to the interface between mini-G$_S$ and A$_{2A}$R in the X-ray structure (PDB code 5g53). The interface in the cryo-EM structure has a buried surface of 1135 Å$^2$ compared to 1048 Å$^2$ for the X-ray structure 5g53; the slight increase is due to interactions between ICL1 of A$_{2A}$R (residues Leu110 and Asn113) and the N-terminal helix of mini-G$_S$ (residues His41 and Arg38). The near full length N-terminal helix was present in the mini-G$_S$ construct in the cryo-EM structure, because this is required for the stable interaction between the α subunit and the βγ subunits, whereas it was truncated and disordered in the X-ray structure. The main interactions between A$_{2A}$R and mini-G$_S$ in both the cryo-EM and X-ray structures are made predominantly by the C-terminal α5 helix in mini-G$_S$ and amino acid residues in H3, H5, H6, H7, H8 and ICL2 of A$_{2A}$R (*Carpenter et al., 2016*). The amino acid residues that make these interactions are identical, but the rotamers sometimes differ between the cryo-EM and X-ray structures. This may be a reflection of the different chemical environments in

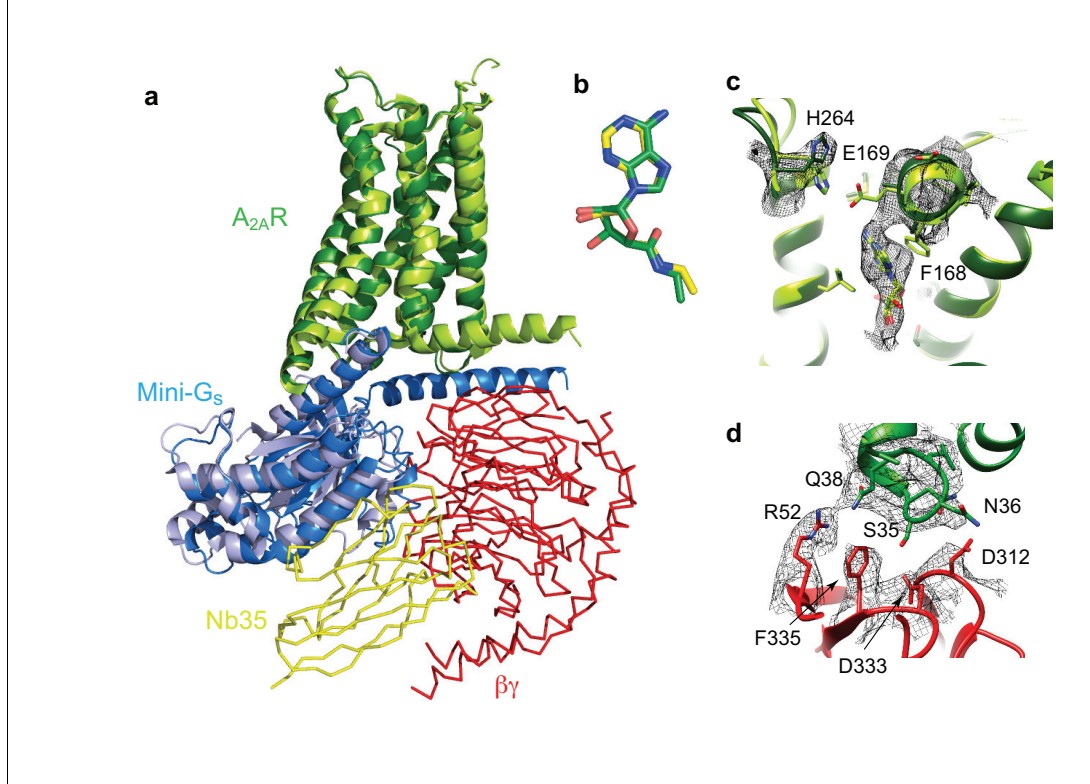

**Figure 6.** Structure of A$_{2A}$R–heterotrimeric G$_S$. (**a**) Superposition of A$_{2A}$R (pale green) coupled to mini-Gs (pale blue) with A$_{2A}$R (dark green) coupled to mini-G$_S$ (dark blue), βγ (red) and Nb35 (yellow). (**b**) Superposition of NECA bound to A$_{2A}$R in the cryo-EM and X-ray structures after alignment of A$_{2A}$R (PyMol). (**c**) The position of His264 in the cryo-EM structure (dark green, density shown by black mesh), differs from its position in the X-ray structure (light green). No density is observed for the side chain of Glu169 in the cryo-EM structure, but when modelled it would be too far away to make a contact with His264. (**d**) The interface between ICL1 of A$_{2A}$R (dark green) and the β subunit (red) is depicted, with density shown as a black mesh.
DOI: https://doi.org/10.7554/eLife.35946.009

which the structures were determined or the slight difference in curvature of the α5 helix in mini-G$_S$. In addition, some interactions may be transient and are captured in one structure and not another. For example, Arg291 at the intracellular end of H7 of A$_{2A}$R adopts a different conformation in the cryo-EM structure compared to the crystal structure. This results in the absence of interactions between the Arg291 side chain and mini-G$_S$, although the backbone carbonyl can still makes potential interactions with Glu392 and the adjacent residues in H8 are still sufficiently close to mini-G$_S$ to make interactions. This region is also the main difference to the β$_2$AR-G$_S$ complex where the α5 helix in G$_S$ does not interact with H7 and H8 of the receptor (*Rasmussen et al., 2011*).

The major difference between the structure determined by cryo-EM of the A$_{2A}$R-heterotrimeric G protein complex and the X-ray structure of the A$_{2A}$R-mini-G$_S$ complex was the presence of the βγ subunit in the cryo-EM structure. No interactions were observed between A$_{2A}$R and the γ subunit, but potential interactions were observed between ICL1 (residues Ser35, Asn36 and Gln38) of A$_{2A}$R and the β subunit (Arg52, Asp312, Asp333 and Phe335). This interface between A$_{2A}$R and the β subunit is considerably more extensive than that observed in the β$_2$AR complex (*Figure 7*), where interactions occur exclusively at Asp312. However, ICL1 shows higher B-factors than the rest of the A$_{2A}$R cryo-EM structure, which may suggest that the interaction is fairly weak. The recently reported cryo-EM structures of the Class B Calcitonin receptor (*Liang et al., 2017*) and Glucagon-like peptide-1 receptor (*Zhang et al., 2017*; *Liang et al., 2018*) bound to heterotrimerc G$_S$ also show a similar interaction between ICL1 and the β subunit where only Asp312 in the β subunit apparently interacts with ICL1 of the receptor. Therefore, the larger interface between the β subunit and A$_{2A}$R appears at the moment unique to this receptor, although the physiological implications are unclear.

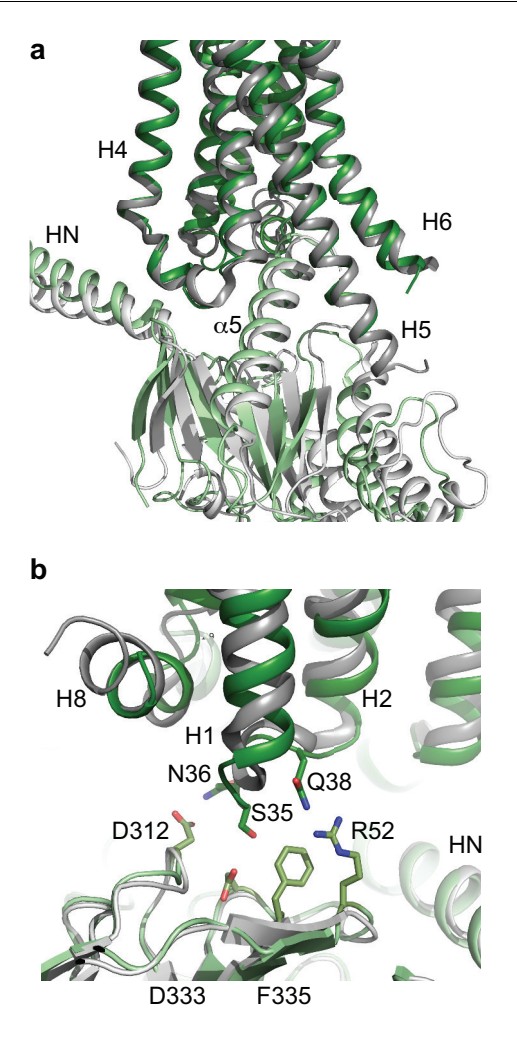

**Figure 7.** Comparison of $A_{2A}R$ and $\beta_2AR$ coupled to heterotrimeric $G_S$. (a) $A_{2A}R$ (dark green) and $\beta_2AR$ (dark grey) were aligned using regions of the receptors predicted to be within the cytoplasmic leaflet of the lipid bilayer. The position of mini-$G_S$ (pale green) coupled to $A_{2A}R$ is compared to the position of the GTPase domain of the $\alpha$ subunit (pale grey) coupled to $\beta_2AR$. The $\beta\gamma$ subunits and Nb35 have been omitted for clarity. (b) Transmembrane region H1 and ECL1 in $A_{2A}R$ (dark green) extends closer to the $\beta$ subunit (pale green), whereas $\beta_2AR$ (dark grey) is too far away from the $\beta$ subunit (pale grey) to make extensive contacts.
DOI: https://doi.org/10.7554/eLife.35946.010

## Discussion

The structure determination of $A_{2A}R$ in complex with mini-$G_S$, $\beta$, $\gamma$ and Nb35 at a physiologically relevant pH has highlighted a number of differences to the structure determined by X-ray crystallography of $A_{2A}R$ coupled to mini-$G_S$ (*Carpenter et al., 2016*). Firstly, contacts between $A_{2A}R$ and the heterotrimeric G protein were identified between ICL1 and the $\beta$ subunit, and between ICL2 and part of the N-terminal $\alpha$-helix of the $\alpha$ subunit; these regions of the G protein were either absent or disordered, respectively, in the crystal structure. Secondly, the difference in pH under which the cryo-EM structure was determined (pH 7.5) compared to many X-ray structures (pH <6) led to ECL2 being more dynamic, as the potential salt bridge between His264 and Glu169 was absent, and consequently Phe168 was disordered. The implications of these observations are discussed more below.

There are now two Class A receptors whose structures have been determined coupled to heterotrimeric $G_S$, $\beta_2AR$ (*Rasmussen et al., 2011*) and $A_{2A}R$, and after this work was completed, two Class B structures coupled to $G_S$ were also published (*Liang et al., 2017*; *Zhang et al., 2017*; *Liang et al., 2018*). As expected the overall architecture of the receptors coupled to $G_S$ are conserved, but the details differ. The biggest difference between coupling of Class A receptors to Class B receptors is that the position of H8 in the Class B receptors is angled towards the G protein by ~30°compared to the Class A receptors. This results in extensive contacts between H8 and the G protein $\beta$ subunit that are absent in Class A receptors. All the receptor structures coupled to $G_S$ show the majority of the contacts between the $\alpha5$ helix of the $\alpha$ subunit and H3, H5 and H6 of the receptor, with receptor-dependent contacts in H2, H7 and H8. The differences may arise partially from the subtle difference in bending of the C-terminal part of the $\alpha5$ helix and the different positions of the $\alpha5$ helix within the receptor, both presumably arising from the different amino acid sequences of the respective receptors. The interactions observed here between $A_{2A}R$ and the $\beta$ subunit are also observed in the Class B receptors, but are absent from the crystal structure of the $\beta_2AR$-$G_S$ structure, although a shift of the $\beta$ subunit by only a few ångstroms would be sufficient for interactions to occur.

The poor density of ECL2 in the cryo-EM map of $A_{2A}R$ coupled to the heterotrimeric G protein, suggests that this region is more dynamic than suggested from the X-ray structures, maybe due to a pH effect that breaks the salt bridge between His264 and Glu169. This salt bridge has been suggested to be highly important in modulating the kinetics of ligand binding (*Segala et al., 2016*). Interestingly, a recent structure (*Sun et al., 2017*) of $A_{2A}R$ bound to compound-1 was crystallised at

pH 6.5, and no crystal contacts were formed by ECL2. In this structure, the N-terminal section of ECL2 lacked density as we observed in the cryo-EM map and, in the latter region, Phe168 adopts two conformations. In one conformation, Phe168 stacks against the ligand and in the other conformation Phe168 points towards the extracellular surface (*Figure 6*). The ionic bridge between Glu169 and His264 seems to be present in this structure, so the two conformations of Phe168 may be a consequence of the ligand. The cryo-EM structure thus adds support to the contention that ECL2 is flexible and may be important in modulating the accessibility of the orthosteric binding site to ligands.

The cryo-EM structure presented here allows for the first time a direct comparison of the structure of a GPCR bound to an identical ligand in the same conformation as determined by cryo-EM and X-ray crystallography. This is highly interesting with respect to drug discovery where tractability and speed of the structure determination are balanced by the resolution required for a particular aspect of any given project. Cryo-EM offers a relatively fast route to the structure of a GPCR in an active conformation coupled to the heterotrimeric G protein $G_S$. The quality of most of the cryo-EM map was very similar to the electron density map from the X-ray structure, despite the reported resolutions being 4.1 Å and 3.4 Å, respectively. This highlights the importance of the local resolution vs the global resolution in cryo-EM maps. In both cases, the ligand density was unambiguous, but adenosine is an asymmetric molecule and difficulties would have been encountered if the ligand was more symmetrical, such as caffeine. However, there is no doubt that cryo-EM is preferred in terms of overall speed; extensive protein engineering is required to obtain crystals of GPCRs (*Tate and Schertler, 2009*), through adding fusion proteins, deletions of flexible regions, removal of post-translational modifications and thermostabilisation. In theory, none of these modifications will be required for a cryo-EM structure, particularly as mild detergents, amphipols and nanodiscs are all compatible with structure determination of membrane proteins by cryo-EM and will maintain often quite unstable membrane proteins in a functional state (*Tate, 2010*). However, once a crystal structure has been determined, they can often attain much higher resolution than structures of membrane proteins obtained so far, although high-resolution cryo-EM structures are possible from single molecule imaging (*Bartesaghi et al., 2015*). Another current advantage of X-ray crystallography is the possibility of soaking crystals to get multiple structures of a receptor bound to different ligands through molecular replacement (*Rucktooa et al., 2018*). Finally, there is still a size limitation of the molecule imaged by cryo-EM for structure determination (*Henderson, 1995*) and experimentally this is now at about 65 kDa (*Khoshouei et al., 2017*). However, given the continued drive towards improving the technology of cryo-EM, there is no doubt that this technique will play a pivotal role in structure-based drug design in future years (*Vinothkumar and Henderson, 2016*).

## Materials and methods

### Expression and purification of the human adenosine A$_{2A}$ receptor

Construction of the thioredoxin-A$_{2A}$R fusion protein and C-terminally truncated A$_{2A}$R (1-317), both containing the N154A mutation, is described elsewhere (*Nehmé et al., 2017*). The constructs were expressed using the baculovirus expression system as described previously (*Carpenter et al., 2016*; *Carpenter and Tate, 2017a*). Cells were harvested by centrifugation 72 hr post-infection, resuspended in hypotonic buffer (20 mM HEPES pH 7.5, 1 mM EDTA, 1 mM PMSF, cOmplete (Roche) protease inhibitor cocktail), flash-frozen in liquid nitrogen and stored at –80°C until use. The purification of the thioredoxin-A$_{2A}$R fusion protein was performed in the detergent LMNG in the presence of 100 µM NECA using Ni$^{2+}$-affinity chromatography followed by SEC as described previously (*Carpenter et al., 2016*; *Carpenter and Tate, 2017a*).

### Preparation of mini-G$_s$ heterotrimer

The mini-G$_S$ construct (399) used in single particle cryo-EM reconstructions is based on the construct 393 that was used in the structure determination of the A$_{2A}$R- mini-G$_S$ crystal structure (*Carpenter et al., 2016*; *Carpenter and Tate, 2016*). However, unlike construct 393, mini-G$_S$399 binds βγ (*Nehmé et al., 2017*). The expression and purification of the respective components and assembly to make the complex containing mini-G$_S$-β$_1$γ$_2$, and the preparation of nanobody Nb35, were all performed following the protocols described previously (*Carpenter and Tate, 2016*; *Rasmussen et al., 2011*; *Carpenter and Tate, 2017b*).

## Preparation of the A$_{2A}$R-mini-G$_S$β$_1$γ$_2$-Nb35 complex

Thioredoxin-A$_{2A}$R, mini-G$_S$-β$_1$γ$_2$ and Nb35 were mixed in a molar ratio of 1:2:4, to yield a final thioredoxin-A$_{2A}$R concentration of 1 mg/ml. 0.1 U of apyrase was added and the mixture was incubated overnight at 4°C. Excess G protein and nanobody were removed by SEC on a Superdex 200 Increase column (running buffer 20 mM HEPES pH 7.5, 100 mM NaCl, 0.1% LMNG, 100 µM NECA). Peak fractions with an absorbance value at 280 nm of 1.5–2 were used immediately for grid preparation or flash frozen in liquid nitrogen and stored at –80°C until use.

## Radioligand binding assays

Insect cells expressing A$_{2A}$R were resuspended in 1 ml of assay buffer (25 mM HEPES pH 7.5, 100 mM KCl, 1 mM MgCl$_2$, protease inhibitor cocktail) at a final concentration of $3 \times 10^6$ cells/ml. Cells were sheared by 10 passages through a bent 26G syringe needle. Cell membranes were diluted 50-fold to 100-fold in assay buffer and aliquots prepared as appropriate. In saturation binding assays, cell membranes containing A$_{2A}$R were incubated with $^3$H-ZM241385 (0.1–40 nM) for 2 hr at 21°C. Non-specific binding was determined in the presence of 10 µM unlabelled ZM241385. In competition binding assays, cell membranes were incubated with NECA (1 nM - 1 µM) for 2 hr at 21°C, in the presence or absence of 25 µM mini-G$_S$393. 5 nM $^3$H-ZM241385 was added followed by a 2 hr incubation. Assays were terminated by filtering through PEI-treated 96-well glass fibre GF/B filter plates (Merck Millipore, Ireland) and washing with ice-cold assay buffer. Filters were dried, placed into scintillation vials and incubated overnight in 4 ml Ultima Gold scintillant (Perkin Elmer). Radioactivity was quantified by scintillation counting using a Tri-Carb counter (Perkin Elmer). Apparent K$_D$ and apparent K$_i$ values were determined using GraphPad Prism version 6.0 (GraphPad Software, San Diego, CA).

## Cryo-EM grid preparation and data collection

Cryo-EM grids were prepared by applying 3 µl of sample (total protein concentration 1 mg/ml) on glow discharged holey gold grids (Quantifoil Au 1.2/1.3 300 mesh). Excess sample was removed by blotting with filter paper for 4–5 s prior to plunge-freezing in liquid ethane using a FEI Vitrobot Mark IV at 100% humidity and 4°C. In all cases, data was collected on a FEI Titan Krios microscope at 300kV. Data without VPP and initial VPP images were acquired using a Gatan K2-Summit detector and a GIF-quantum energy filter (Gatan) with a 20 eV slit and zero loss mode to remove inelastic scattering. For the initial non-VPP dataset, EPU automatic data collection software (FEI) was used while the VPP date set of the K2-summit detector was collected using SerialEM automatic data collection software (*Mastronarde, 2005*).

The non-VPP data set contained a total of 2800 micrographs, collected as 40 movie frames at a dose rate of 2 e$^-$/pixel/sec (1.25 e$^-$/Å$^2$ per frame) for 10 s, with a total accumulated dose of 50 e$^-$/Å$^2$. The magnification was 200,000x yielding 0.66 Å/pixel at the specimen level.

The K2-VPP dataset was the result of merging three datasets with slightly different collection parameters: (a) 213 micrographs collected as 40 movie frames at 5.2 e$^-$/pixel/s over 10 s for a total dose of 40 e$^-$/Å$^2$; (b) 232 micrographs collected as 23 frames at 9 e$^-$/pixel/s over 4.6 s for a total accumulated dose of 30 e$^-$/Å$^2$; (c) 461 micrographs collected as 30 movie frames at a dose rate of 6 e$^-$/pixel/s over 6.5 s for a total dose of 30 e$^-$/Å$^2$. In all cases the magnification was set to obtain a pixel size of 1.14 Å.

One data set was acquired using a Falcon III detector in electron counting mode by recording 75 movie frames (0.8 s per frame) at a dose rate of 0.5 e$^-$/pixel/s (0.4 e$^-$/Å$^2$ per frame) for a total accumulated dose of 30 e$^-$/Å$^2$ acquired over a period of 60 s. Pixel size at the specimen was calibrated to be 1.07 Å. A total of 827 images were incorporated into the dataset.

## Data processing and model building

All data processing were performed using RELION-2 (*Kimanius et al., 2016*). Good quality images were selected manually and drift correction, beam induced motion and dose weighting was performed for each of the datasets with MotionCor2 (*Zheng et al., 2017*), using $5 \times 5$ patches and the corresponding dose per frame. CTF fitting and phase shift estimation were performed using Gctf-v0.1.06 (*Zhang, 2016*). In all cases, auto-picking (*Scheres, 2015*) was performed with a Gaussian blob as a template (*Fernandez-Leiro and Scheres, 2017*). Elimination of false positives or 'bad

particles' was performed over two rounds of reference-free 2D classification. 10,000 random particles were used for *ab initio* model generation using the Stochastic Deepest Descent (SDG) algorithm incorporated in RELION-2.1. The resulting model was used as input for the initial 3D classification. After a single round of 3D classification, particles in quality models were pooled together for refinement. The Falcon III-VPP data was divided into three classes, where two of them presented clear structural features resembling a GPCR-G-protein heterotrimer complex. During refinement of the Falcon III + VPP data, the low-pass filter effect of the Wiener filter in the regularised likelihood optimisation algorithm was relaxed through the use of a regularisation parameter (T = 5). This allowed the refinement algorithm to consider higher spatial frequencies in the alignment of the individual particles yielding a map of higher quality. Nevertheless, both half-reconstructions were kept completely separately, and the final resolution estimate (at the post-processing stage in RELION) was based on the standard FSC between the two unfiltered half-reconstructions. Signal subtraction of the micelle was performed as described in the results section and were used only in the last iteration of refinement. Application of 'particle polishing' in RELION (corrects for beam induced motion and performs experimental dose-weighting) did not improve the quality of the density. Local resolution was calculated with RELION.

Model building and refinement was carried out using the CCP-EM software suite (*Burnley et al., 2017*). The activated $A_{2A}R$ and mini-$G_S$ coordinates were taken as starting models (PDB code 5g53) together with the βγ coordinated from the $β_2AR$ complex structure (*Rasmussen et al., 2011*). Jelly-body refinement was performed in REFMAC5 (*Murshudov et al., 2011*) followed by manual modification and real space refinement in Coot (*Emsley and Cowtan, 2004*). Refinement with restraints (generated in ProSMART [*Nicholls et al., 2012*]) was performed in REFMAC5 in order to maintain the secondary structure in regions with poorer map quality. Validation of the model was performed in Coot, Molprobity (*Chen et al., 2010*) and EMRinger (*Barad et al., 2015*). The goodness of fit of the model to the map was carried out using Phenix (*Adams et al., 2010*), using a global model-vs-map FSC correlation (*Figure 3*).

## Acknowledgements

This work was funded by a grant from the European Research Council (EMPSI 339995) Heptares Therapeutics Ltd, Pfizer and core funding from the Medical Research Council [MRC U105197215]. We thank Rishi Matadeen and Kasim Sader for their help with cryo-EM data collection, Christos Savva for his help in using the VPP and Wim Hagen for the SerialEM single-particle data collection script. We also thank Rafael Fernandez-Leiro, Sjors Scheres and Paula da Fonseca for useful discussions.

## Additional information

### Competing interests

Christopher G Tate: is a consultant and shareholder of Heptares Therapeutics, and they also funded this work. The other authors declare that no competing interests exist.

### Funding

| Funder | Grant reference number | Author |
| --- | --- | --- |
| Medical Research Council | U105197215 | Christopher G Tate |
| European Research Council | EMPSI 339995 | Christopher G Tate |
| Heptares Therapeutics | | Christopher G Tate |
| Pfizer UK | | Christopher G Tate |

The funders had no role in study design, data collection and interpretation, or the decision to submit the work for publication.

## Author contributions
Javier García-Nafría, Data curation, Formal analysis, Validation, Investigation, Visualization, Methodology, Writing—original draft, Writing—review and editing; Yang Lee, Data curation, Formal analysis, Validation, Investigation, Methodology, Writing—original draft, Writing—review and editing; Xiaochen Bai, Formal analysis, Investigation, Methodology; Byron Carpenter, Resources; Christopher G Tate, Conceptualization, Supervision, Funding acquisition, Writing—original draft, Project administration, Writing—review and editing

## Author ORCIDs
Byron Carpenter https://orcid.org/0000-0003-1712-3528
Christopher G Tate http://orcid.org/0000-0002-2008-9183

## Decision letter and Author response
Decision letter https://doi.org/10.7554/eLife.35946.019
Author response https://doi.org/10.7554/eLife.35946.020

# Additional files

## Supplementary files
• Transparent reporting form
DOI: https://doi.org/10.7554/eLife.35946.011

## Data availability
Structural data have been deposited in the PDB under the accession code 6gdg and in EMDB with accession code 4390

The following datasets were generated:

| Author(s) | Year | Dataset title | Dataset URL | Database, license, and accessibility information |
|---|---|---|---|---|
| Garcia-Nafria J, Lee Y | 2018 | Cryo-EM structure of the adenosine A2A receptor bound to a miniGs heterotrimer | https://www.rcsb.org/structure/6gdg | Publicly available at the RCSB Protein Data Bank (accession no. 6GDG) |
| Javier García-Nafría, Yang Lee, Xiaochen Bai, Byron Carpenter, Christopher G Tate | 2018 | Cryo-EM structure of the adenosine A2A receptor bound to a miniGs heterotrimer | http://www.ebi.ac.uk/pdbe/entry/emdb/EMD-4390 | Publicly available at the Electron Microscopy Data Bank (accession no. EMD-4390) |

The following previously published dataset was used:

| Author(s) | Year | Dataset title | Dataset URL | Database, license, and accessibility information |
|---|---|---|---|---|
| Carpenter B, Nehme R, Warne T, Leslie AG, Tate CG | 2016 | Structure of the adenosine A(2A) receptor bound to an engineered G protein. | https://www.rcsb.org/structure/5g53 | Publicly available at the RCSB Protein Data Bank (accession no. 5g53) |

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
