## [Decision Letter]

Thank you for submitting your article "Cryo-EM structure of the adenosine A_2A_ receptor coupled to an engineered heterotrimeric G protein" for consideration by *eLife*. Your article has been favorably evaluated by Richard Aldrich (Senior Editor) and three reviewers, one of whom, Werner Kühlbrandt (Reviewer #1), is a member of our Board of Reviewing Editors. The following individual involved in review of your submission has agreed to reveal their identity: Patrick Sexton (Reviewer #3).

The reviewers have discussed the reviews with one another and the Reviewing Editor has drafted this decision to help you prepare a revised submission.

Summary:

This is a follow-up study of work by the same group published in 2016, where they characterized the adenosine A_2A_ receptor bound to an engineered α subunit of G_S_ protein. Here they solve the structure of A_2A_R in complex with an engineered heterotrimeric G_S_ protein, including the βγ subunits, by using cryo-EM. This allows for comparison of these two structures, that reveals differences in the ligand binding pocket and G protein-binding interface. Additionally, the authors present a practical protocol of cryo-EM for structure determination of GPCRs. Overall, the methodology is well described and the text is clearly written.

The main interest of the manuscript is the comparison of the structure determined by x-ray crystallography and by cryoEM with or without a Volta phase plate, with two different cameras. The study is timely and will be interesting and useful for the growing number of researchers engaged in the structure determination of membrane protein complexes by cryo-EM.

Suggested revisions:

1) The manuscript compares the performance of the K2 camera against the Falcon III (which comes out very well), and of the K2 camera with and without phase plate. Both comparisons are very instructive. My only regret is that the analysis does not extend to a comparison of the Falcon III camera with and without phase plate. This would make the paper even more interesting and useful, but a request to include it would not be reasonable. It would be good however to know how much difference the phase plate really makes. Might it not be possible, with the better Falcon III camera, carefully optimised samples (thin ice) and the same strategy that improved the resolution of the Falcon III/VPP structure from 4.5 to 4.1 Å (!), to collect more particles and make do without the phase plate? The VPP is not understood, poorly controlled, temperamental and its use can be tricky even for experienced cryo-EM practitioners.

2) Please add an extra line to Table 1, giving the total number of particles used, rather than only the particles contributing to the final map.

3) It would do no harm to place a little more emphasis on a nice point that is made almost in passing: a 4.1 Å cryo-EM structure is as good as or better than a 3.4 Å x-ray structure of the same protein. This will not come as a surprise to cryo-EM aficionados but it will to many protein crystallographers, and it has rarely been made so clearly.

4) One of the concerning issues is the discussion about the interaction between Glu169 and His264. The authors proposed that the absence of this interaction in the cryo-EM structure is due to the deprotonation of His264 at pH 7.5. However, given the low resolution and poor densities in the ECL2 region, modelling into this region is ambiguous. Therefore the presence of this interaction should not be ruled out. Discussion about such detailed side-chain interactions at low resolution is somewhat speculative and should be avoided. In addition, it was mentioned that "this ionic bridge affects small molecule binding kinetics". I assume this is from previous studies. How does this interaction affect the ligand binding? Is it pH dependent? Do these data support the current study? These should be included in the manuscript.

5) There are differences in the G protein binding interface between A_2A_R and β_2A_R. Could the authors comment on this based on the available complex structures? Interactions between receptor ICL1 and G_S_ protein were also observed in the recently published cryo-EM structures of class B GPCRs. Any differences between A_2A_R and class B GPCRs on this? A comment should be included in the manuscript.

6) With respect to the specific new structure solved, comparison to the previously published x-ray structure of the same receptor bound to the mini-G_S_ (without βγ subunits) provides a relatively modest increase in understanding of the A_2A_ receptor, but does identify regions that are likely physiologically dynamic, and of contributions of the βγ-subunits to conformation of the α-subunit and to direct receptor interaction. The significance of these interactions is not experimentally explored. Thus, while this is of interest, it has limited novelty as a new structure.

7) Is the data deposited in appropriate databases for access? Is all the map data being deposited?

8) Please include PDB validation reports for the cryo-EM structures.

---

## [Author Response]

Suggested revisions:1) The manuscript compares the performance of the K2 camera against the Falcon III (which comes out very well), and of the K2 camera with and without phase plate. Both comparisons are very instructive. My only regret is that the analysis does not extend to a comparison of the Falcon III camera with and without phase plate. This would make the paper even more interesting and useful, but a request to include it would not be reasonable. It would be good however to know how much difference the phase plate really makes. Might it not be possible, with the better Falcon III camera, carefully optimised samples (thin ice) and the same strategy that improved the resolution of the Falcon III/VPP structure from 4.5 to 4.1 Å (!), to collect more particles and make do without the phase plate? The VPP is not understood, poorly controlled, temperamental and its use can be tricky even for experienced cryo-EM practitioners.

The data collected using only the K2 versus K2 + VPP were performed as equivalent as possible, so we feel this truly reflects the effect of the VPP on data quality using the K2 detector. Repeating this comparison using the Falcon III would be interesting, however we feel that the improvements produced using the VPP would be comparable. Given the extremely limited time we have on the Titan Krios, we cannot justify the use of scarce resources for this experiment.

2) Please add an extra line to Table 1, giving the total number of particles used, rather than only the particles contributing to the final map.

We have added more information to Table 1 including the total number of particles used after 2D classification (which we believe are the closest to the real number of ‘true’ particles), as well as more details about data collection, model validation and correlation between model and map.

3) It would do no harm to place a little more emphasis on a nice point that is made almost in passing: a 4.1 Å cryo-EM structure is as good as or better than a 3.4 Å x-ray structure of the same protein. This will not come as a surprise to cryo-EM aficionados but it will to many protein crystallographers, and it has rarely been made so clearly.

We feel this is not an entirely fair comparison and could be a misleading statement when made in general. The overall resolution of 4.1 Å for our data encompasses the whole of A_2A_R and the G protein, but the local resolution is far better in the core of the receptor, reaching resolutions of at least 3.8 Å resolution. However, even this is an oversimplification given variations in the shapes of FSC curves and how this can affect the quality of the map. We have already made the point in the Discussion (fourth paragraph) but we do not wish to overemphasise this point any further.

4) One of the concerning issues is the discussion about the interaction between Glu169 and His264. The authors proposed that the absence of this interaction in the cryo-EM structure is due to the deprotonation of His264 at pH 7.5. However, given the low resolution and poor densities in the ECL2 region, modelling into this region is ambiguous. Therefore the presence of this interaction should not be ruled out. Discussion about such detailed side-chain interactions at low resolution is somewhat speculative and should be avoided. In addition, it was mentioned that "this ionic bridge affects small molecule binding kinetics". I assume this is from previous studies. How does this interaction affect the ligand binding? Is it pH dependent? Do these data support the current study? These should be included in the manuscript.

The reference for how this ionic bridge modulated ligand binding kinetics has been added. The poorer density in this region has now been acknowledged in the text. However, the rotamer for the bigger side chain of His264 is confidently placed, and points outwards (see Figure 6C). It should be noted that H264 is in a well-ordered region of ECL3 and not in the less-ordered part of ECL2, where we agree that the positions of side chains is less certain. Although the side chain for Glu169 cannot be modelled, the rotamer for His264 is equivalent to that found in crystal structures where the pH is higher and the ionic bridge is lost. We therefore conclude that the ionic bridge is unlikely to be present in the cryo-EM structure. This has been made clearer in the text. See below:

“A second difference between the NECA-bound X-ray structures and the cryo-EM structure is a likely absence of an interaction between Glu169 and His264. […] Although the rotamer for Glu169 cannot be assigned in the cryo-EM map, His264 adopts a similar rotamer as seen in crystal structures obtained at higher pH, such as the complexes with caffeine, XAC and ZM241385 (~pH 8), all showing a broken ionic bridge.”

5) There are differences in the G protein binding interface between A_2A_R and β_2A_R. Could the authors comment on this based on the available complex structures? Interactions between receptor ICL1 and G_S_ protein were also observed in the recently published cryo-EM structures of class B GPCRs. Any differences between A_2A_R and class B GPCRs on this? A comment should be included in the manuscript.

The major differences in the interaction between the α subunit and β_2_AR vs. A_2A_R have already been extensively discussed in our previous paper on the structure of A_2A_R coupled to mini-G_S_ (Carpenter et al., 2016), so we have not reiterated them here, except for adding an extra sentence highlighting the major difference (subsection “Comparison of the G protein–receptor interface between the crystal and cryo-EM structures”). We had already included the main differences between the β_2_AR-G_S_ complex and the A_2A_R complex in terms of interactions with the β subunit in the aforementioned subsection.

We have also added a comparison of the ICL1-β subunit interactions and the Class B Calcitonin receptor and GLP-1 receptor cryo-EM structures:

“This interface between A_2A_ R and the β subunit is considerably more extensive than that observed in the β_2_AR complex (Figure 7), where interactions occur exclusively at Asp312. [...] Therefore the larger interface between the β subunit and A_2A_ R appears at the moment unique to this receptor, although the physiological implications are unclear.”

*6) With respect to the specific new structure solved, comparison to the previously published x-ray structure of the same receptor bound to the mini-*G_S_*(without βγ subunits) provides a relatively modest increase in understanding of the A_2A_ receptor, but does identify regions that are likely physiologically dynamic, and of contributions of the βγ-subunits to conformation of the α-subunit and to direct receptor interaction. The significance of these interactions is not experimentally explored. Thus, while this is of interest, it has limited novelty as a new structure.*

The physiological significance of the contacts between A_2A_R and the β subunit are indeed not understood. There is a huge amount of effort being directed towards this from a structural perspective and we await further structures with different β subunits to see if there are indeed any differences between them. The role of these contacts will be best determined by extensive mutagenesis coupled with biophysical and in cell coupling assays, but these are beyond the scope of the current manuscript. We hope that others will use our structure to probe into this interesting observation.

7) Is the data deposited in appropriate databases for access? Is all the map data being deposited?

The PDB and the final map have been deposited in the PDB and EMDB and accession numbers have been included in Table 1.

8) Please include PDB validation reports for the cryo-EM structures.

The PDB validation report has been included. Table 1 has been also updated with further validation measures.